# Deep Decoder: Concise Image Representations from Untrained Non-convolutional Networks

**Reinhard Heckel**
Department of Electrical and Computer Engineering
Rice University
rh43@rice.edu

**Paul Hand**
Department of Mathematics and
College of Computer and Information Science
Northeastern University
p.hand@northeastern.edu

## Abstract

Deep neural networks, in particular convolutional neural networks, have become highly effective tools for compressing images and solving inverse problems including denoising, inpainting, and reconstruction from few and noisy measurements. This success can be attributed in part to their ability to represent and generate natural images well. Contrary to classical tools such as wavelets, image-generating deep neural networks have a large number of parameters—typically a multiple of their output dimension—and need to be trained on large datasets. In this paper, we propose an untrained simple image model, called the deep decoder, which is a deep neural network that can generate natural images from very few weight parameters. The deep decoder has a simple architecture with no convolutions and fewer weight parameters than the output dimensionality. This underparameterization enables the deep decoder to compress images into a concise set of network weights, which we show is on par with wavelet-based thresholding. Further, underparameterization provides a barrier to overfitting, allowing the deep decoder to have state-of-the-art performance for denoising. The deep decoder is simple in the sense that each layer has an identical structure that consists of only one upsampling unit, pixel-wise linear combination of channels, ReLU activation, and channelwise normalization. This simplicity makes the network amenable to theoretical analysis, and it sheds light on the aspects of neural networks that enable them to form effective signal representations.

## 1 Introduction

Data models are central for signal and image processing and play a key role in compression and inverse problems such as denoising, super-resolution, and compressive sensing. These data models impose structural assumptions on the signal or image, which are traditionally based on expert knowledge. For example, imposing the assumption that an image can be represented with few non-zero wavelet coefficients enables modern (lossy) image compression (Antonini et al., 1992) and efficient denoising (Donoho, 1995).

In recent years, it has been demonstrated that for a wide range of imaging problems, from compression to denoising, deep neural networks trained on large datasets can often outperform methods based on traditional image models (Toderici et al., 2016; Agustsson et al., 2017; Theis et al., 2017; Burger et al., 2012; Zhang et al., 2017). This success can largely be attributed to the ability of deep networks to represent realistic images when trained on large datasets. Examples include learned representations via autoencoders (Hinton & Salakhutdinov, 2006) and generative adversarial models (Goodfellow et al., 2014). Almost exclusively, three common features of the recent success stories of using deep neural network for imaging related tasks are i) that the corresponding networks are over-parameterized (i.e., they have much more parameters than the dimension of the image that they represent or generate), ii) that the networks have a convolutional structure, and perhaps most importantly, iii) that the networks are trained on large datasets.

An important exception that breaks with the latter feature is a recent work by Ulyanov et al. Ulyanov et al. (2018), which provides an algorithm, called the deep image prior (DIP), based on deep neural networks, that can solve inverse problems well without any training. Specifically, Ulyanov et al. demonstrated that fitting the weights of an over-parameterized deep convolutional network to a single image, together with strong regularization by early stopping of the optimization, performs competitively on a variety of image restoration problems. This result is surprising because it does not involve a training dataset, which means that the notion of what makes an image 'natural' is contained in a combination of the network structure and the regularization. However, without regularization the proposed network has sufficient capacity to overfit to noise, preventing meaningful image denoising.

These prior works demonstrating the effectiveness of deep neural networks for image generation beg the question whether there may be a deep neural network model of natural images that is underparameterized and whose architecture alone, without algorithmic assistance, forms an efficient model for natural images.

In this paper, we propose a simple image model in the form of a deep neural network that can represent natural images well while using very few parameters. This model thus enables image compression, denoising, and solving a variety of inverse problems with close to or state of the art performance. We call the network the deep decoder, due to its resemblance to the decoder part of an autoencoder. The network does not require training, and contrary to previous approaches, the network itself incorporates all assumptions on the data, is under-parameterized, does not involve convolutions, and has a simplicity that makes it amenable to theoretical analysis. The key contributions of this paper are as follows:

- The network is under-parameterized. Thus, the network maps a lower-dimensional space to a higher-dimensional space, similar to classical image representations such as sparse wavelet representations. This feature enables image compression by storing the coefficients of the network after its weights are optimized to fit a single image. In Section 2, we demonstrate that the compression is on-par with wavelet thresholding (Antonini et al., 1992), a strong baseline that underlies JPEG-2000. An additional benefit of underparameterization is that it provides a barrier to overfitting, which enables regularization of inverse problems.

- The network *itself* acts as a natural data model. Not only does the network require no training (just as the DIP Ulyanov et al. (2018)); it also does not critically rely on regularization, for example by early stopping (in contrast to the DIP). The property of not involving learning has at least two benefits: The same network and code is usable for a number of applications, and the method is not sensitive to a potential misfit of training and test data.

- The network does not use convolutions. Instead, the network does have pixelwise linear combinations of channels, and, just like in a convolutional neural network, the weights are shared among spatial positions. Nonetheless, these are not convolutions because they provide no spatial coupling between pixels, despite how pixelwise linear combinations are sometimes called '1x1 convolutions.' In contrast, the majority of the networks for image compression, restoration, and recovery have convolutional layers with filters of nontrivial spatial extent Toderici et al. (2016); Agustsson et al. (2017); Theis et al. (2017); Burger et al. (2012); Zhang et al. (2017). This work shows that relationships characteristic of nearby pixels of natural images can be imposed directly by upsampling layers.

- The network only consists of a simple combination of few building blocks, which makes it amenable to analysis and theory. For example, we prove that the deep decoder can only fit a small proportion of noise, which, combined with the empirical observation that it can represent natural images well, explains its denoising performance.

The remainder of the paper is organized as follows. In Section 2, we first demonstrate that the deep decoder enables concise image representations. We formally introduce the deep decoder in Section 3. In Section 4, we show the performance of the deep decoder on a number of inverse problems such as denoising. In Section 5 we discuss related work, and finally, in Section 6 we provide theory and explanations on what makes the deep decoder work.

## 2 CONCISE IMAGE REPRESENTATIONS WITH A DEEP IMAGE MODEL

Intuitively, a model describes a class of signals well if it is able to represent or approximate a member of the class with few parameters. In this section, we demonstrate that the deep decoder, an untrained, non-convolutional neural network, defined in the next section, enables *concise* representation of an image—on par with state of the art wavelet thresholding.

The deep decoder is a deep image model $G \colon \mathbb{R}^N \to \mathbb{R}^n$, where $N$ is the number of parameters of the model, and $n$ is the output dimension, which is (much) larger than the number of parameters ($n \gg N$). The parameters of the model, which we denote by $\mathbf{C}$, are the weights of the network, and not the input of the network, which we will keep fixed. To demonstrate that the deep decoder enables concise image representations, we choose the number of parameters of the deep decoder, $N$, such that it is a small fraction of the output dimension of the deep decoder, i.e., the dimension of the images[1].

We draw 100 images from the ImageNet validation set uniformly at random and crop the center to obtain a 512x512 pixel color image. For each image $\mathbf{x}^*$, we fit a deep decoder model $G(\mathbf{C})$ by minimizing the loss

$$L(\mathbf{C}) = \|G(\mathbf{C}) - \mathbf{x}^*\|_2^2$$

with respect to the network parameters $\mathbf{C}$ using the Adam optimizer. We then compute for each image the corresponding peak-signal-to-noise ratio, defined as $10 \log_{10}(1/\mathrm{MSE})$, where $\mathrm{MSE} = \frac{1}{3 \cdot 512^2} \|\mathbf{x}^* - G(\mathbf{C})\|_2^2$, $G(\mathbf{C})$ is the image generated by the network, and $\mathbf{x}^*$ is the original image.

We compare the compression performance to wavelet compression (Antonini et al., 1992) by representing each image with the $N$-largest wavelet coefficients. Wavelets—which underly JPEG 2000, a standard for image compression—are one of the best methods to approximate images with few coefficients. In Fig. 1 we depict the results. It can be seen that for large compression factors ($3 \cdot 512^2/N = 32.3$), the representation by the deep decoder is slightly better for most images (i.e., is above the red line), while for smalle compression factors ($3 \cdot 512^2/N = 8$), the wavelet representation is slightly better. This experiment shows that deep neural networks can represent natural images well with very few parameters and without any learning.

The observation that, for small compression factors, wavelets enable more concise representations than the deep decoder is intuitive because any image can be represented exactly with sufficiently many wavelet coefficients. In contrast, there is no reason to believe a priori that the deep decoder has zero representation error because it is underparameterized.

The main point of this experiment is to demonstrate that the deep decoder is a good image model, which enables applications like solving inverse problems, as in Section 4. However, it also suggest that the deep decoder can be used for lossy image compression, by quantizing the coefficients $\mathbf{C}$ and saving the quantized coefficients. In the appendix, we show that image representations of the deep decoder are not sensitive to perturbations of its coefficients, thus quantization does not have a detrimental effect on the image quality. Deep networks were used successfully before for the compression of images (Toderici et al., 2016; Agustsson et al., 2017; Theis et al., 2017). In contrast to our work, which is capable of compressing images without any learning, the aforementioned works *learn* an encoder and decoder using convolutional recurrent neural networks (Toderici et al., 2016) and convolutional autoencoders (Theis et al., 2017) based on training data.

## 3 THE DEEP DECODER

We consider a decoder architecture that transforms a randomly chosen and fixed tensor $\mathbf{B}_0 \in \mathbb{R}^{n_0 \times k_0}$ consisting of $k_0$ many $n_0$-dimensional channels to an $n_d \times k_{\mathrm{out}}$ dimensional image, where $k_{\mathrm{out}} = 1$ for a grayscale image, and $k_{\mathrm{out}} = 3$ for an RGB image with three color channels. Throughout, $n_i$ has two dimensions; for example our default configuration has $n_0 = 16 \times 16$ and $n_d = 512 \times 512$. The network transforms the tensor $\mathbf{B}_0$ to an image by pixel-wise linearly combining the channels, upsampling operations, applying rectified linear units (ReLUs), and normalizing

---

[1]Specifically, using notation defined in Section 3, we took a deep decoder $G$ with $d = 6$ layers and output dimension $512 \times 512 \times 3$, and choose $k = 64$ and $k = 128$ for the large and small compression factors, respectively.

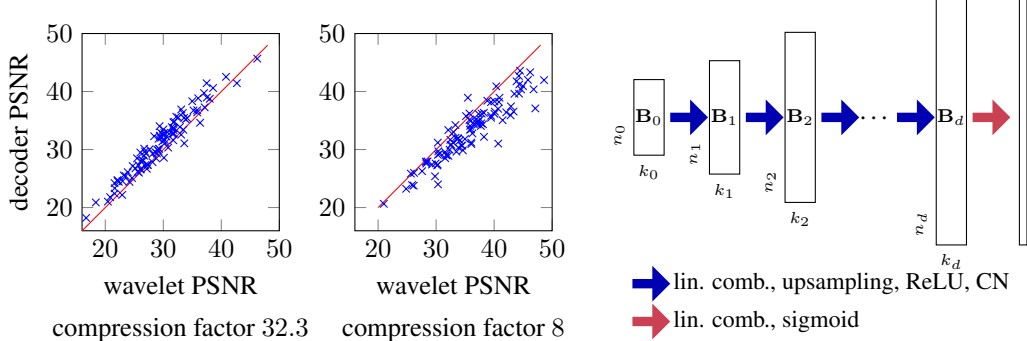

Figure 1: The deep decoder (depicted on the right) enables concise image representations, on-par with state-of-the-art wavelet based compression. The crosses on the left depict the PSNRs for 100 randomly chosen ImageNet-images represented with few wavelet coefficients and with a deep decoder with an equal number of parameters. A cross above the red line means the corresponding image has a smaller representation error when represented with the deep decoder. The deep decoder is particularly simple, as each layer has the same structure, consisting of a pixel-wise linear combination of channels, upsampling, ReLU nonlinearities, and channelwise normalization (CN).

the channels. Specifically, the channels in the $(i + 1)$-th layer are given by

$$\mathbf{B}_{i+1} = \mathrm{cn}(\mathrm{relu}(\mathbf{U}_i \mathbf{B}_i \mathbf{C}_i)), \quad i = 0, \ldots, d - 1.$$

Here, the coefficient matrices $\mathbf{C}_i \in \mathbb{R}^{k_i \times k_{i+1}}$ contain the weights of the network. Each column of the tensor $\mathbf{B}_i \mathbf{C}_i \in \mathbb{R}^{n_i \times k_{i+1}}$ is formed by taking linear combinations of the channels of the tensor $\mathbf{B}_i$ in a way that is consistent across all pixels.

Then, $\mathrm{cn}(\cdot)$ performs a channel normalization operation which is equivalent to normalizing each channel individually, and can be viewed as a special case of the popular batch normalization proposed in (Ioffe & Szegedy, 2015). Specifically, let $\mathbf{Z}_i = \mathrm{relu}(\mathbf{U}_i \mathbf{B}_i \mathbf{C}_i)$ be the channels in the $i$-th layer, and let $\mathbf{z}_{ij}$ be the $j$-th channel in the $i$-th layer. Then channel normalization performs the following transformation: $\mathbf{z}'_{ij} = \frac{\mathbf{z}_{ij} - \mathrm{mean}(\mathbf{z}_{ij})}{\sqrt{\mathrm{var}(\mathbf{z}_{ij}) + \epsilon}} \gamma_{ij} + \beta_{ij}$, where mean and var compute the empirical mean and variance, and $\gamma_{ij}$ and $\beta_{ij}$ are parameters, learned independently for each channel, and $\epsilon$ is a fixed small constant. Learning the parameter $\gamma$ and $\beta$ helps the optimization but is not critical. This is a special case of batch normalization with batch size one proposed in (Ioffe & Szegedy, 2015), and significantly improves the fitting of the model, just like how batch norm alleviates problems encountered when training deep neural networks.

The operator $\mathbf{U}_i \in \mathbb{R}^{n_{i+1} \times n_i}$ is an upsampling tensor, which we choose throughout so that it performs bi-linear upsampling. For example, if the channels in the input have dimensions $n_0 = 16 \times 16$, then the upsampling operator $\mathbf{U}_0$ upsamples each channel to dimensions $32 \times 32$. In the last layer, we do not upsample, which is to say that we choose the corresponding upsampling operator as the identity. Finally, the output of the $d$-layer network is formed as

$$\mathbf{x} = \mathrm{sigmoid}(\mathbf{B}_d \mathbf{C}_d),$$

where $\mathbf{C}_d \in \mathbb{R}^{k_d \times k_{\mathrm{out}}}$. See Fig. 1 for an illustration. Throughout, our default architecture is a $d = 6$ layer network with $k_i = k$ for all $i$, and we focus on output images of dimensions $n_d = 512 \times 512$ and number of channels $k_{\mathrm{out}} = 3$. Recall that the parameters of the network are given by $\mathbf{C} = \{\mathbf{C}_0, \mathbf{C}_1, \ldots, \mathbf{C}_d\}$, and the output of the network is only a function of $\mathbf{C}$, since we choose the tensor $\mathbf{B}_0$ at random and fix it. Therefore, we write $\mathbf{x} = G(\mathbf{C})$. Note that the number of parameters is given by $N = \sum_{i=1}^{d}(k_i k_{i+1} + 2k_i) + k_{\mathrm{out}} k_d$ where the term $2k_i$ corresponds to the two free parameters associated with the channel normalization. Thus, the number of parameters is $N = dk^2 + 2dk + 3k$. In the default architectures with $d = 6$ and $k = 64$ or $k = 128$, we have that $N = 25{,}536$ (for $k = 64$) and $N = 100{,}224$ ($k = 128$) out of an RGB image space of dimensionality $512 \times 512 \times 3 = 786{,}432$ parameters.

We finally note that naturally variations of the deep decoder are possible; for example in a previous version of this manuscript, we applied upsampling after applying the relu-nonlinearity, but found that applying it before yields slightly better results.

## 3.1 A NON-CONVOLUTIONAL NETWORK?

While the deep decoder does not use convolutions, its structure is closely related to that of a convolutional neural network. Specifically, the network does have pixelwise linear combinations of channels, and just like in a convolutional neural network, the weights are shared among spatial positions. Nonetheless, pixelwise linear combinations are not proper convolutions because they provide no spatial coupling of pixels, despite how they are sometimes called $1 \times 1$ convolutions. In the deep decoder, the source of spatial coupling is only from upsampling operations.

In contrast, a large number of networks for image compression, restoration, and recovery have convolutional layers with filters of nontrivial spatial extent Toderici et al. (2016); Agustsson et al. (2017); Theis et al. (2017); Burger et al. (2012); Zhang et al. (2017). Thus, it is natural to ask whether using linear combinations as we do, instead of actual convolutions yields better results.

Our simulations indicate that, indeed, linear combinations yield more concise representations of natural images than $p \times p$ convolutions, albeit not by a huge factor. Recall that the number of parameters of the deep decoder with $d$ layers, $k$ channels at each layer, and $1 \times 1$ convolutions is $N(d, k; 1) = dk^2 + 3k + 2dk$. If we consider a deep decoder with convolutional layers with filters of size $p \times p$, then the number of parameters is: $N(d, k; p) = p^2(dk^2 + 3k) + 2dk$. If we fix the number of channels, $k$, but increase $p$ to 3, the representation error only decreases since we increase the number of parameters (by a factor of approximately $3^2$). We consider image reconstruction as described in Section 2. For a meaningful comparison, we keep the number of parameters fixed, and compare the representation error of a deep decoder with $p = 1$ and $k = 64$ (the default architecture in our paper) to a variant of the deep decoder with $p = 3$ and $k = 22$, so that the number of parameters is essentially the same in both configurations. We find that the representation of the deep decoder with $p = 1$ is better (by about 1dB, depending on the image), and thus for concise image representations, linear combinations ($1 \times 1$ convolutions) appear to be more effective than convolutions of larger spatial extent.

## 4 SOLVING INVERSE PROBLEMS WITH THE DEEP DECODER

In this section, we use the deep decoder as a structure-enforcing model or regularizers for solving standard inverse problems: denoising, super-resolution, and inpainting. In all of those inverse problems, the goal is to recover an image $\mathbf{x}$ from a noisy observation $\mathbf{y} = f(\mathbf{x}) + \eta$. Here, $f$ is a known forward operator (possibly equal to identity), and $\eta$ is structured or unstructured noise.

We recover the image $\mathbf{x}$ with the deep decoder as follows. Motivated by the finding from the previous section that a natural image $\mathbf{x}$ can (approximately) be represented with the deep decoder as $G(\mathbf{C})$, we estimate the unknown image from the noisy observation $\mathbf{y}$ by minimizing the loss

$$L(\mathbf{C}) = \|f(G(\mathbf{C})) - \mathbf{y}\|_2^2$$

with respect to the model parameters $\mathbf{C}$. Let $\hat{\mathbf{C}}$ be the result of the optimization procedure. We estimate the image as $\hat{\mathbf{x}} = G(\hat{\mathbf{C}})$.

We use the Adam optimizer for minimizing the loss, but have obtained comparable results with gradient descent. Note that this optimization problem is non-convex and we might not reach a global minimum. Throughout, we consider the least-squares loss (i.e., we take $\|\cdot\|_2$ to be the $\ell_2$ norm), but the loss function can be adapted to account for structure of the noise.

We remark that fitting an image model to observations in order to solve an inverse problem is a standard approach and is not specific to the deep decoder or deep-network-based models in general. Specifically, a number of classical signal recovery approaches fit into this framework; for example solving a compressive sensing problem with $\ell_1$-norm minimization amounts to choosing the forward operator as $f(\mathbf{x}) = \mathbf{A}\mathbf{x}$ and minimizing over $\mathbf{x}$ in a $\ell_1$-norm ball.

### 4.1 DENOISING

We start with the perhaps most basic inverse problem, denoising. The motivation to study denoising is at least threefold: First, denoising is an important problem in practice, second, many inverse problem can be solved as a chain of denoising steps (Romano et al., 2017), and third, the denoising

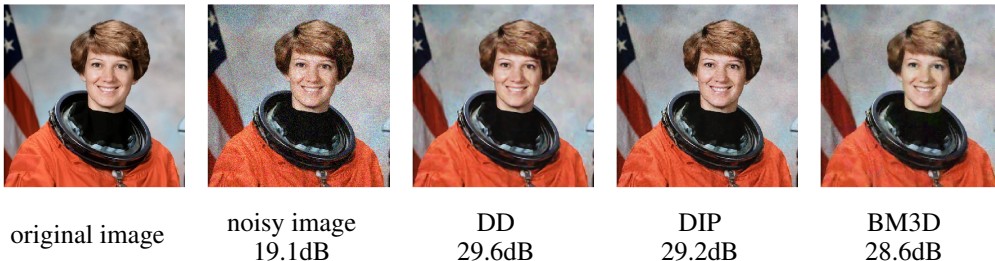

| original image | noisy image
19.1dB | DD
29.6dB | DIP
29.2dB | BM3D
28.6dB |

Figure 2: An application of the deep decoder for denoising the astronaut test image. The deep decoder has performance on-par with state of the art untrained denoising methods, such as the DIP method (Ulyanov et al., 2018) and the BM3D algorithm (Dabov et al., 2007).

problem is simple to model mathematically, and thus a common entry point for gaining intuition on a new method. Given a noisy observation $\mathbf{y} = \mathbf{x} + \eta$, where $\eta$ is additive noise, we estimate an image with the deep decoder by minimizing the least squares loss $\|G(\mathbf{C}) - \mathbf{y}\|_2^2$, as described above.

The results in Fig. 2 and Table 1 demonstrate that the deep decoder has denoising performance on-par with state of the art untrained denoising methods, such as the related Deep Image Prior (DIP) method (Ulyanov et al., 2018) (discussed in more detail later) and the BM3D algorithm (Dabov et al., 2007). Since the deep decoder is an untrained method, we only compared to other state-of-the-art untrained methods (as opposed to learned methods such as (Zhang et al., 2017)).

Why does the deep decoder denoise well? In a nutshell, from Section 2 we know that the deep decoder can represent natural images well even when highly underparametrized. In addition, as a consequence of being under-parameterized, the deep decoder can only represent a small proportion of the noise, as we show analytically in Section 6, and as demonstrated experimentally in Fig. 4. Thus, the deep decoder "filters out" a significant proportion of the noise, and retains most of the signal.

How to choose the parameters of the deep decoder? The larger $k$, the larger the number of latent parameters and thus the smaller the representation error, i.e., the error that the deep decoder makes when representing a noise-free image. On the other hand, the smaller $k$, the fewer parameters, and the smaller the range space of the deep decoder $G(\mathbf{C})$, and thus the more noise the method will remove. The optimal $k$ trades off those two errors; larger noise levels require smaller values of $k$ (or some other form of regularization). If the noise is significantly larger, then the method requires either choosing $k$ smaller, or it requires another means of regularization, for example early stopping of the optimization. For example $k = 64$ or $128$ performs best out of $\{32, 64, 128\}$, for a PSNR of around 20dB, while for a PSNR of about 14dB, $k = 32$ performs best.

### 4.2 SUPERRESOLUTION

We next super-resolve images with the deep denoiser. We define a forward model $f$ that performs downsampling with the Lanczos filter by a factor of four. We then downsample a given image by a factor of four, and then reconstruct it with the deep decoder (with $k = 128$, as before). We compare performance to bi-cubic interpolation and to the deep image prior, and find that the deep decoder outperforms bicubic interpolation, and is on-par with the deep image prior (see Table 1 in the appendix).

### 4.3 INPAINTING

Finally, we use the deep decoder for inpainting, where we are given an inpainted image $\mathbf{y}$, and a forward model $f$ mapping a clean image to an inpainted image. The forward model $f$ is defined by a mask that describes the inpainted region, and simply maps that part of the image to zero. Fig. 3 and Table 1 demonstrate that the deep decoder performs well on the inpainting problems; however, the deep image prior performs slightly better on average over the examples considered. For the impainting problem we choose a significantly more expressive prior, specifically $k = 320$.

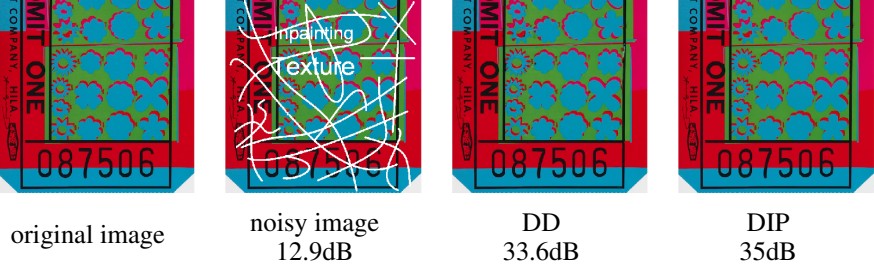

| original image | noisy image | DD | DIP |
|:---:|:---:|:---:|:---:|
| | 12.9dB | 33.6dB | 35dB |

Figure 3: An application of the deep decoder for recovering an inpainted image. For this example, the deep decoder and the deep image perform almost equally well.

## 5 RELATED WORK

Image compression, restoration, and recovery algorithms are either trained or untrained. Conceptually, the deep decoder image model is most related to untrained methods, such as sparse representations in overcomplete dictionaries (for example wavelets (Donoho, 1995) and curvelets (Starck et al., 2002)). A number of highly successful image restoration and recovery schemes are not directly based on generative image models, but rely on structural assumptions about the image, such as exploiting self-similarity in images for denoising (Dabov et al., 2007) and super-resolution (Glasner et al., 2009).

Since the deep decoder is an image-generating deep network, it is also related to methods that rely on trained deep image models. Deep learning based methods are either trained end-to-end for tasks ranging from compression (Toderici et al., 2016; Agustsson et al., 2017; Theis et al., 2017; Burger et al., 2012; Zhang et al., 2017) to denoising (Burger et al., 2012; Zhang et al., 2017), or are based on learning a generative image model (by training an autoencoder or GAN (Hinton & Salakhutdinov, 2006; Goodfellow et al., 2014)) and then using the resulting model to solve inverse problems such as compressed sensing (Bora et al., 2017; Hand & Voroninski, 2018), denoising (Heckel et al., 2018), phase retrieval (Hand et al., 2018; Shamshad & Ahmed, 2018), and blind deconvolution (Asim et al., 2018), by minimizing an associated loss. In contrast to the deep decoder, where the optimization is over the weights of the network, in all the aforementioned methods, the weights are adjusted only during training and then are fixed upon solving the inverse problem.

Most related to our work is the Deep Image Prior (DIP), recently proposed by Ulyanov et al. (Ulyanov et al., 2018). The deep image prior is an untrained method that uses a network with an hourglass or encoder-decoder architecture, similar to the U-net and related architectures that work well as autoencoders. The key differences to the deep decoder are threefold: i) the DIP is over-parameterized, whereas the deep decoder is under-parameterized. ii) Since the DIP is highly over-parameterized, it critically relies on regularization through early stopping and adding noise to its input, whereas the deep decoder does not need to be regularized (however, regularization can enhance performance). iii) The DIP is a convolutional neural network, whereas the deep decoder is not.

We further illustrate point ii) comparing the DIP and deep decoder by denoising the astronaut image from Fig. 2. In Fig. 4(a) we plot the Mean Squared Error (MSE) over the number of iterations of the optimizer for fitting the noisy astronaut image $\mathbf{x} + \eta$. Note that to fit the model, we minimize the error $\|G(\mathbf{C}) - (\mathbf{x} + \eta)\|_2^2$, because we are only given the noisy image, but we plot the MSE between the representation and the actual, true image $\|G(\mathbf{C}^t) - \mathbf{x}\|_2^2$ at iteration $t$. Here, $\mathbf{C}^t$ are the parameters of the deep decoder after $t$ iterations of the optimizer. In Fig. 4(b) and (c), we plot the loss or MSE associated with fitting the noiseless astronaut image, $\mathbf{x}$ ($\|G(\mathbf{C}^t) - \mathbf{x}\|_2^2$) and the noise itself, $\eta$, ($\|G(\mathbf{C}^t) - \eta\|_2^2$). Models are fitted independently for the noisy image, the noiseless image, and the noise.

The plots in Fig. 4 show that with sufficiently many iterations, both the DIP and the DD can fit the image well. However, even with a large number of iterations, the deep decoder can not fit the noise well, whereas the DIP can. This is not surprising, given that the DIP is over-parameterized and the deep decoder is under-parameterized. In fact, in Section 6 we formally show that due to the

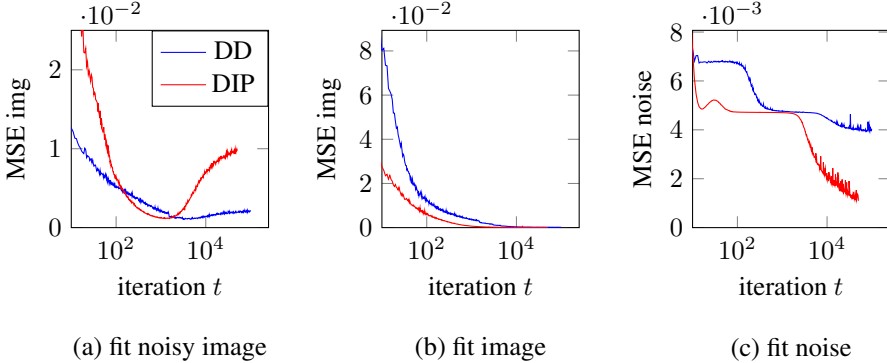

(a) fit noisy image  (b) fit image  (c) fit noise

Figure 4: Denoising with the deep decoder and the deep image prior. The first two panels shows the MSE of the output of the DD or DIP for a noisy or noiseless image relative to the noiseless image. The third panel shows the MSE of the output of DD or DIP for an image consisting purely of noise, as computed relative to that noise. Due to under-parameterization, the deep decoder can only fit a small proportion of the noise, and thus enables image denoising. Early stopping can mildly enhance the performance of DD; to see this note that in panel (a), the minimum is obtained at around 5000 iterations and not at 50,000. The deep image prior can fit noise very well, but fits an image faster than noise, thus early stopping is critical for denoising performance.

underparameterization, the deep decoder can only fit a small proportion of the noise, no matter how and how long we optimize. As a consequence, it filters out much of the noise when applied to a natural image. In contrast, the DIP relies on the empirical observation that the DIP fits a structured image faster than it fits noise, and thus critically relies on early stopping.

## 6    DISCUSSION ON WHAT MAKES THE DECODER WORK

In the previous sections we empirically showed that the deep decoder can represent images well and at the same time cannot fit noise well. In this section, we formally show that the deep decoder can only fit a small proportion of the noise, relative to the degree of underparameterization. In addition, we provide insights into how the components of the deep decoder contribute to representing natural images well, and we provide empirical observations on the sensitivity of the parameters and their distribution.

### 6.1    THE DEEP DECODER CAN ONLY FIT LITTLE NOISE

We start by showing that an under-parameterized deep decoder can only fit a proportion of the noise relative to the degree of underparameterization. At the heart of our argument is the intuition that a method mapping from a low- to a high-dimensional space can only fit a proportion of the noise relative to the number of free parameters. For simplicity, we consider a one-layer network, and ignore the batch normalization operation. Then, the networks output is given by

$$G(\mathbf{C}) = \mathrm{relu}(\mathbf{U}_0\mathbf{B}_0\mathbf{C}_0)\mathbf{c}_1 \in \mathbb{R}^n.$$

Here, we take $\mathbf{C} = (\mathbf{C}_0, \mathbf{c}_1)$, where $\mathbf{C}_0$ is a $k \times k$ matrix and $\mathbf{c}_1$ is a $k$-dimensional vector, assuming that the number of output channels is 1. While for the performance of the deep decoder the choice of upsampling matrix is important, it is not relevant for showing that the deep decoder cannot represent noise well. Therefore, the following statement makes no assumptions about the upsampling matrix $\mathbf{U}_0$.

**Proposition 1.** *Consider a deep decoder with one layer and arbitrary upsampling and input matrices. That is, let $\mathbf{B}_0 \in \mathbb{R}^{n_0 \times k}$ and $\mathbf{U}_0 \in \mathbb{R}^{n \times n_0}$. Let $\eta \in \mathbb{R}^n$ be zero-mean Gaussian noise with covariance matrix $\sigma\mathbf{I}$, $\sigma > 0$. Assume that $k^2 \log(n_0)/n \leq 1/32$. Then, with probability at least $1 - 2n_0^{-k^2}$,*

$$\min_{\mathbf{C}} \|G(\mathbf{C}) - \eta\|_2^2 \geq \|\eta\|_2^2 \left(1 - 20\frac{k^2 \log(n_0)}{n}\right).$$

The proposition asserts that the deep decoder can only fit a small portion of the noise energy, precisely a proportion determined by its number of parameters relative to the output dimension, $n$. Our

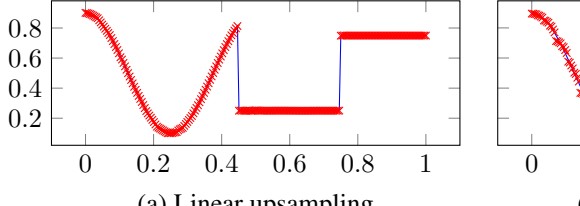 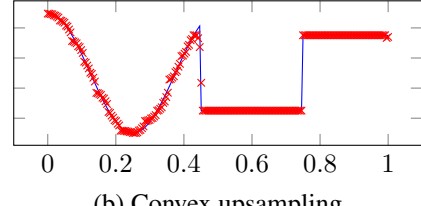

| (a) Linear upsampling | (b) Convex upsampling |
| --- | --- |

Figure 5: The blue curves show a one-dimensional piecewise smooth signal, and the red crosses show estimates of this signal by a one-dimensional deep decoder with either linear or convex upsampling. We see that linear upsampling acts as an indirect signal prior that promotes piecewise smoothness.

simulations and preliminary analytic results suggest that this statement extends to multiple layers in that the lower bound becomes $\left(1 - c\frac{k^2 \log(\prod_{i=1}^{d} n_{i-1})}{n}\right)$, where $c$ is a numerical constant. Note that the lower bound does not directly depend on the noise variance $\sigma$ since both sides of the inequality scale with $\sigma^2$.

## 6.2 UPSAMPLING

Upsampling is a vital part of the deep decoder because it is the only way that the notion of locality explicitly enters the signal model. In contrast, most convolutional neural networks have spatial coupling between pixels both by unlearned upsampling, but also by learned convolutional filters of nontrivial spatial extent. The choice of the upsampling method in the deep decoder strongly affects the 'character' of the resulting signal estimates. We now discuss the impacts of a few choices of upsampling matrices $\mathbf{U}_i$, and their impact on the images the model can fit.

**No upsampling:** If there is no upsampling, or, equivalently, if $\mathbf{U}_i = \mathbf{I}$, then there is no notion of locality in the resulting image. All pixels become decoupled, and there is then no notion of which pixels are near to each other. Specifically, a permutation of the input pixels (the rows of $\mathbf{B}_0$) simply induces the identical permutation of the output pixels. Thus, if a deep decoder without upsampling could fit a given image, it would also be able to fit random permutations of the image equally well, which is practically equivalent to fitting random noise.

**Nearest neighbor upsampling:** If the upsampling operations perform nearest neighbor upsampling, then the output of the deep decoder consists of piecewise constant patches. If the upsampling doubles the image dimensions at each layer, this would result in patches of $2^d \times 2^d$ pixels that are constant. While this upsampling method does induce a notion of locality, it does so too strongly in the sense that squares of nearby pixels become identical and incapable of fitting local variation within natural images.

**Linear and convex, non-linear upsampling:** The specific choice of upsampling matrix affects the multiscale 'character' of the signal estimates. To illustrate this, Figure 5 shows the signal estimate from a 1-dimensional deep decoder with upsampling operations given by linear upsampling $(x_0, x_1, x_2, \ldots) \mapsto (x_0, 0.5x_0 + 0.5x_1, x_1, 0.5x_1 + 0.5x_2, x_2, \ldots)$ and convex nonlinear upsampling given by $(x_0, x_1, x_2, \ldots) \mapsto (x_0, 0.75x_0 + 0.25x_1, x_1, 0.75x_1 + 0.25x_2, x_2, \ldots)$. Note that while both models are able to capture the coarse signal structure, the convex upsampling results in a multiscale fractal-like structure that impedes signal representation. In contrast, linear upsampling is better able to represent smoothly varying portions of the signal. Linear upsampling in a deep decoder indirectly encodes the prior that natural signals are piecewise smooth and in some sense have approximately linear behavior at multiple scales

## 6.3 NETWORK INPUT

Throughout, the network input is fixed. We choose the network input $\mathbf{B}_1$ by choosing its entries uniformly at random. The particular choice of the input is not very important; it is however desirable that the rows are incoherent. To see this, as an extreme case, if any two rows of $\mathbf{B}_1$ are equal and if the upsampling operation preserves the values of those pixels exactly (for example, as with the linear upsampling from the previous section), then the corresponding pixels of the output image is

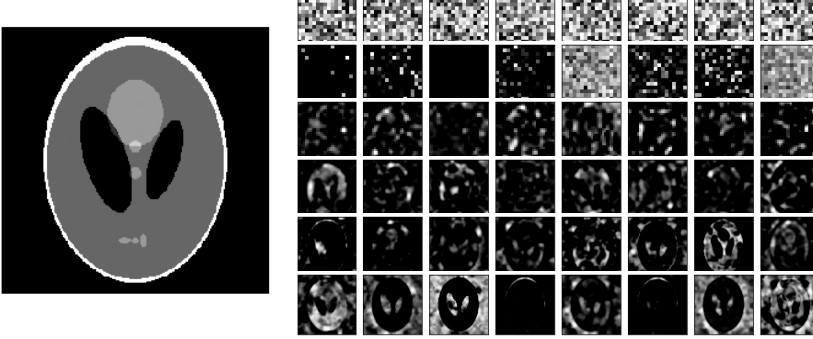

Figure 6: The left panel shows an image reconstruction after training a deep decoder on the MRI phantom image (PSNR is 51dB). The right panel shows how the deep decoder builds up an image starting from a random input. From top to bottom are the input to the network and the activation maps (i.e., $\mathrm{relu}(\mathbf{B}_i\mathbf{C}_i)$) for eight out of the $64$ channels in layers one to six.

|    |          | barbara | lovett | mri  | zebra | F16  | baboon | fruit | astronaut | castle | saturn |
|----|----------|---------|--------|------|-------|------|--------|-------|-----------|--------|--------|
| DN | identity | 20.3    | 20.9   | 22.1 | 21.3  | 20.3 | 20.3   | 20.5  | 20.6      | 20.4   | 20.2   |
|    | DD128    | **26.8**| **27.9**| 26.9 | 22.5  | **29.1**| 21.4 | **29.2**| **29.8**| **27.7**| 29.0  |
|    | DIP      | 24.4    | 25.3   | 26.6 | **24.8**| 25.0 | **22.8**| 25.7 | 26.1      | 25.0   | 25.0   |
|    | BM3D     | 24.7    | 25.1   | **28.0**| 22.8 | 25.2 | 22.6   | 26.3  | 26.2      | 25.6   | **30.5**|
| SR | bicubic  | 26.3    | 26.0   | 24.5 | 18.2  | 26.4 | **20.7**| 27.1 | 29.3      | 25.8   | 27.9   |
|    | DD128    | **26.4**| 26.3   | **26.4**| 19.0 | 26.6 | 20.6   | **28.6**| 30.2    | **26.1**| 27.8  |
|    | DIP      | 26.4    | **26.6**| 25.6 | **19.2**| **27.4**| 20.6 | 28.3 | 29.6      | 26.0   | **27.9**|
| IP | identity | 14.9    | 14.4   | 18.3 | 13.0  | 11.7 | 14.0   | 12.4  | 14.0      | 14.2   | 13.4   |
|    | DD320    | 32.3    | **33.6**| 31.4 | **24.4**| **34.9**| 24.9 | **36.6**| **36.5**| 32.5 | **36.7**|
|    | DIP      | **35.6**| 26.9   | **32.1**| 24.2 | 34.7 | **26.2**| 35.5 | 35.3      | **32.6**| 36.2  |

Table 1: Performance comparison of the deep decoder for denoising (DN), superresolution (SR), and inpainting (IP), in peak signal to noise ratio (PSNR). Note that identity corresponds to the PSNR of the noise and corruption in the DN and IP experiments, respectively.

also exactly the same, which restricts the range space of the deep decoder unrealistically, since for any pair of pixels, the majority of natural images does not have exactly the same value at this pair of pixels.

### 6.4 Image generation by successive approximation

The deep decoder is tasked with coverting multiple noise channels into a structured signal primarily using pixelwise linear combinations, ReLU activation funcions, and upsampling. Using these tools, the deep decoder builds up an image through a series of successive approximations that gradually morph between random noise and signal. To illustrate that, we plot the activation maps (i.e., $\mathrm{relu}(\mathbf{B}_i\mathbf{C}_i)$) of a deep decoder fitted to the phantom MRI test image (see Fig. 6). We choose a deep decoder with $d = 5$ layers and $k = 64$ channels. This image reconstruction approach is in contrast to being a semantically meaningful hierarchical representation (i.e., where edges get combined into corners, that get combined into simple sample, and then into more complicated shapes), similar to what is common in discriminative networks.

### Acknowledgments

RH is partially supported by NSF award IIS-1816986, an NVIDIA Academic GPU Grant, and would like to thank Ludwig Schmidt for helpful discussions on the deep decoder in general, and in particular for suggestions on the experiments in Section 2.

Code to reproduce the results is available at `https://github.com/reinhardh/supplement_deep_decoder`

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

APPENDIX

## A    PROOF OF PROPOSITION 1

Suppose that the network has one layer, i.e., $G(\mathbf{C}) = \mathrm{relu}(\mathbf{U}_0\mathbf{B}_0\mathbf{C}_0)\mathbf{c}_1$. We start by re-writing $\mathbf{B}_1 = \mathrm{relu}(\mathbf{B}_0\mathbf{C}_0)$ in a convenient form. For a given vector $\mathbf{x} \in \mathbb{R}^n$, denote by $\mathrm{diag}(\mathbf{x} > 0)$ the matrix that contains one on its diagonal if the respective entry of $\mathbf{x}$ is positive and zero otherwise. Let $\mathbf{c}_{jci}$ denote the $i$-th column of $\mathbf{C}_j$, and denote by $\mathbf{W}_{ji} \in \{0, 1\}^{k \times k}$ the corresponding diagonal matrix $\mathbf{W}_{ji} = \mathrm{diag}(\mathbf{U}_j\mathbf{B}_j\mathbf{c}_{jci} > 0)$. With this notation, we can write
$$\mathbf{B}_1 = \mathrm{relu}(\mathbf{U}_0\mathbf{B}_0\mathbf{C}_0) = [\mathbf{W}_{01}\mathbf{U}_0\mathbf{B}_0\mathbf{c}_{0c1}, \dots, \mathbf{W}_{0k}\mathbf{U}_0\mathbf{B}_0\mathbf{c}_{0ck}].$$
Thus,
$$G(\mathbf{C}) = [\mathbf{W}_{01}\mathbf{U}_0\mathbf{B}_0, \dots, \mathbf{W}_{0k}\mathbf{U}_0\mathbf{B}_0] \begin{bmatrix} \mathbf{c}_{0c1}[\mathbf{c}_1]_1 \\ \vdots \\ \mathbf{c}_{0c1}[\mathbf{c}_1]_k \end{bmatrix},$$
where $[\mathbf{c}_1]_i$ denotes the $i$-th entry of $\mathbf{c}_1$. Thus, $G(\mathbf{C})$ lies in the union of at-most-$k^2$-dimensional subspaces of $\mathbb{R}^n$, where each subspace is determined by the matrices $\{\mathbf{W}_{0j}\}_{j=1}^k$. The number of those subspaces is bounded by $n^{k^2}$. This follows from the fact that for the matrix $\mathbf{A} := \mathbf{U}_0\mathbf{B}_0$, by Lemma 1 below, the number of different matrices $\mathbf{W}_{0j}$ is bounded by $n^k$. Since there are $k$ matrices, the number of different sets of matrices is bounded by $n^{k^2}$.

**Lemma 1.** *For any* $\mathbf{A} \in \mathbb{R}^{n \times k}$ *and* $k \geq 5$,
$$|\{\mathrm{diag}(\mathbf{A}\mathbf{v} > 0)\mathbf{A} | \mathbf{v} \in \mathbb{R}^k\}| \leq n^k.$$

Next, fix the matrixes $\{\mathbf{W}_{0j}\}_j$. As $G(\mathbf{C})$ lies in an at-most-$k^2$-dimensional subspace, let $S$ be a $k^2$-dimensional subspace that contains the range of $G$ for these fixed $\{\mathbf{W}_{0j}\}_j$. It follows that
$$\min_{\mathbf{C}} \|G(\mathbf{C}) - \eta\|_2^2 \geq \frac{\|P_{S^c}\eta\|_2^2}{\|\eta\|_2^2}. \tag{1}$$

Now, we make use of the following bound on the projection of the noise $\eta$ onto a subspace.

**Lemma 2.** *Let* $S \subset \mathbb{R}^n$ *be a subspace with dimension* $\ell$. *Let* $\eta \sim \mathcal{N}(0, I_n)$ *and* $\beta \geq 1$. *Then,*
$$\mathrm{P}\left[\frac{\|P_{S^c}\eta\|_2^2}{\|\eta\|_2^2} \geq 1 - \frac{10\beta\ell}{n}\right] \geq 1 - e^{-\beta\ell} - e^{-n/16}.$$

*Proof of Lemma 2.* From Laurent & Massart (2000, Lem. 1), if $X \sim \chi_n^2$, then
$$\mathrm{P}\left[X - n \geq 2\sqrt{nx} + 2x\right] \leq e^{-x},$$
$$\mathrm{P}\left[X \leq n - 2\sqrt{nx}\right] \leq e^{-x}.$$
With these, we obtain
$$\mathrm{P}[X \geq 5\beta n] \leq e^{-\beta n} \text{ if } \beta \geq 1, \tag{2}$$
$$\mathrm{P}[X \leq n/2] \leq e^{-n/16}. \tag{3}$$
We have $\frac{\|P_{S^c}\eta\|_2^2}{\|\eta\|_2^2} = 1 - \frac{\|P_S\eta\|_2^2}{\|\eta\|_2^2}$. Note that $\|P_S\eta\|_2 \sim \chi_\ell^2$ and $\|\eta\|_2^2 \sim \chi_n^2$. Applying inequality (2) to bound $\|P_S\eta\|_2$ and inequality (3) to bound $\|\eta\|_2^2$, a union bound gives that claim. $\square$

Thus, by inequality (1) and Lemma 2 with $\ell = k^2$, for all $\beta \geq 1$,
$$\mathrm{P}\left[\frac{1}{\|\eta\|_2^2} \min_{\mathbf{C}} \|G(\mathbf{C}) - \eta\|_2^2 \geq 1 - \frac{10\beta k^2}{n} \bigg| \{\mathbf{W}_{0j}\}_j\right] \geq 1 - e^{-k^2\beta} - e^{-n/16}. \tag{4}$$

Since the number of matrices $\{\mathbf{W}_{0j}\}_j$ is bounded by $n_0^{k^2}$, by a union bound,
$$\mathrm{P}\left[\frac{1}{\|\eta\|_2^2} \min_{\mathbf{C}} \|G(\mathbf{C}) - \eta\|_2^2 \leq 1 - \frac{10\beta k^2}{n}\right] \leq n_0^{k^2}(e^{-\beta k^2} + e^{-n/16}) \leq 2n_0^{-k^2}, \tag{5}$$
where the last inequality follows with choosing $\beta = 2\log(n_0)$ and by the assumption that $k^2 < \frac{n}{32\log n_0}$. This proves the claim in Proposition 1.

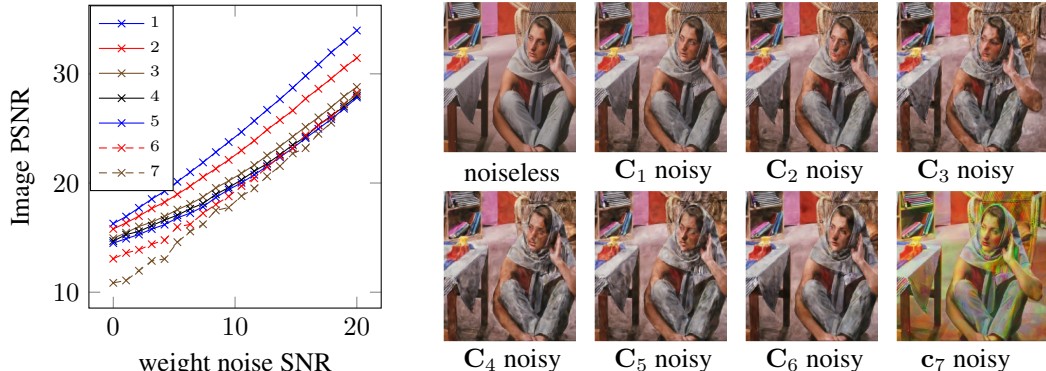

Figure 7: Sensitivity to parameter perturbations of the weights in each layer, and images generated by perturbing the weights in different layers, and keeping the weights in the other layers constant.

## A.1 PROOF OF LEMMA 1

Our goal is to count the number of sign patterns $(\mathbf{A}\mathbf{v} > 0) \in \{0, 1\}$. Note that this number is equal to the maximum number of partitions one can get when cutting a $k$-dimensional space with $n$ many hyperplanes that all pass through the origin, and are perpendicular to the rows of $\mathbf{A}$. This number if well known (see for example Winder (1966)) and is upper bounded by

$$2 \sum_{i=0}^{n-1} \binom{n-1}{k}.$$

Thus,

$$|\{\operatorname{diag}(\mathbf{A}\mathbf{v} > 0)\mathbf{A} \colon \mathbf{v} \in \mathbb{R}^k\}| \leq 2 \sum_{i=0}^{n-1} \binom{n-1}{k} \leq 2k \left(\frac{e(n-1)}{k}\right)^k \leq n^k,$$

where the last inequality holds for $k \geq 5$.

## B SENSITIVITY TO PARAMETER PERTURBATIONS AND DISTRIBUTION OF PARAMETERS

The deep decoder is not overly sensitive to perturbations of its coefficients. To demonstrate this, fit the standard test image Barbara with a deep decoder with 6 layers and $k = 128$, as before. We then perturb the weights in a given layer $i$ (i.e., the matrix $\mathbf{C}_i$) with Gaussian noise of a certain signal-to-noise ratio relative to $\mathbf{C}_i$ and leave the other weights and the input untouched. We then measure the peak signal-to-noise ratio in the image domain, and plot the corresponding curve for each layer (see Fig. 7). It can be seen that the representation provided by the deep decoder is relatively stable with respect to perturbations of its coefficients, and that it is more sensitive to perturbations in higher levels.

Finally, in Fig. 8 we depict the distribution of the weights of the network after fitted to the Barbara test image, and note that the weights are approximately Gaussian distributed.

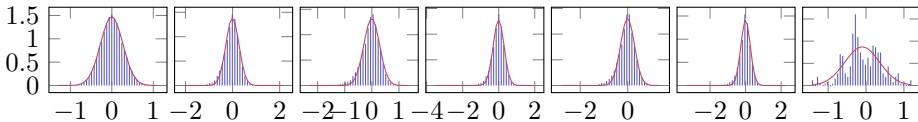

Figure 8: Distribution of the weights for fitting the test image Barbara along with a Gaussian fit: The distribution of the weighs is approximately Gaussian.

