# OpenReview forum: "Deep Decoder: Concise Image Representations from Untrained Non-convolutional Networks"
_ICLR.cc/2019/Conference_

### Official Review · AnonReviewer1 · 2018-10-24
**Overall a nice paper**

**Rating:** 7
**Confidence:** 3

**Review:**

The paper builds upon Deep Image Prior (DIP) - work which shows that one can optimize a neural generator to fit a single image without learning on any dataset, and the output of the generator (which approximates the image) can be used for denoising / super resolution / etc. The paper proposes a new architecture for the DIP method which has much less parameters, but works on par with DIP. Another contribution of the paper is theoretical treatment of (a simplified version of) the proposed architecture showing that it can’t fit random noise (and thus maybe better suited for denoising).

The paper is clearly written, and the proposed architecture has too cool properties: it’s compact enough to be used for image compression; and it doesn’t overfit thus making early stopping notnesesary (which was crucial for the original DIP model).

I have two main concerns about this paper.
First, it is somewhat misleading about its contributions: it's not obvious from abstract/introduction that the whole model is the same as DIP except for the proposed architecture. Specifically, the first contribution listed in the introduction makes it look like this paper introduces the idea of not learning the decoder on the dataset (the one that starts with “The network is not learned and itself incorporates all assumptions on the data.”).

My second concern is about the theoretical contribution. On the one hand, I enjoyed the angle the authors tackled proving that the network architecture is underparameterized enough to be a good model for denoising. On the other hand, the obtained results are very weak: only one layered version of the paper is analysed and the theorem applies only to networks with less than some threshold of parameters. Roughly, the theorem states that if for example we fix any matrix B of size e.g. 256 x k and matrix U of size 512 x 256 and then compute U relu(B C) where C is the vector of parameters of size k x 1, AND if k < 2.5 (i.e. if we use at most 2 parameters), then it would be very hard to fit 512 iid gaussian values (i.e. min_C ||U relu(B C) - eta|| where eta ~ N(0, 1)). This restriction of the number of parameters to be small is only mentioned in the theorem itself, not in the discussion of its implications.
Also, the theorem only applies to the iid noise, while most natural noise patterns have structure (e.g. JPEG artifacts, broken pixels, etc) and thus can probably be better approximated with deep models.

Since the paper manages to use very few parameters (BTW, how many parameters in total do you have? Can you please add this number to the text?), it would be cool to see if second order methods like LBFGS can be applied here.

Some less important points:

Fig 4 is very confusing.
First, it doesn’t label the X axis.
Second, the caption mentions that early stopping is beneficial for the proposed method, but I can’t see it from the figure.
Third, I don’t get what is plotted on different subplots. The text mentions that (a) is fitting the noisy image, (b) is fitting the noiseless image, and (c) is fitting noise. Is it all done independently with three different models? Then why does the figure says test and train loss? And why DIP loss goes up, it should be able to fit anything, right? If not and it’s a single model that gets fitted on the noisy image and tested on the noiseless image, then how can you estimate the level of noise fitting? ||G(C) - eta|| should be high if G(C) ~= x.
Also, in this quote “In Fig. 4(a) we plot the Mean Squared Error (MSE) over the number of iterations of the optimizer for fitting the noisy astronaut image x + η (i.e., FORMULA ...” the formula doesn’t correspond to the text.
And finally, the discussion of this figure makes claims about the behaviour of the model that seems to be too strong to be based on a single image experiment.

I don’t get the details of the batch normalization used: with respect to which axis the mean and variance are computed?

The authors claim that the model is not convolutional. But first, it’s not obvious why this would be a good thing (or a bad thing for that matter). Second, it’s not exactly correct (as noted in the paper itself): the architecture uses 1x1 convolutions and upsampling, which combined give a weak and underparametrized analog of convolutions.

> The deep decoder is a deep image model G: R N → R n, where N is the number of parameters of the model, and n is the output dimension, which is typically much larger than the number of parameters (N << n).
I think it should be vice versa, N >> n

The following footnote
> Specifically, we took a deep decoder G with d = 6 layers and output dimension 512×512×3, and choose k = 64 and k = 128 for the respective compression ratios.
Uses unintroduced (at that point) notation and is very confusing.

It would be nice to have a version of Figure 6 with k = 6, so that one can see all feature maps (in contrast to a subset of them).

I’m also wondering, is it harder to optimize the proposed architecture compared to DIP? The literature on distillation indicates that overparameterization can be beneficial for convergence and final performance.

---

> ### Author Response · Authors · 2018-11-13
> **response**
>
> Many thanks for the detailed review!
>
> Main comments:
> 1/ The DIP approach critically relies on regularization in order to make the method work (both by adding random noise in each optimization step to the input, as well as early stopping).
> As the first reviewer noted ``In fact, the DIP of Ulyanov et al. can hardly be considered "a model" (or a prior, for that matter), and instead should be considered "an algorithm", since it relies on the early stopping of a specific optimization algorithm''.
>
> However we follow the reviewers' suggestion and made clear that the idea to use a deep network without learning as an image model is not new and rewrote the item to ``The network itself acts as a natural data model.  Not only does the network require no training (just as the DIP); it also does not critically rely on regularization, for example by early stopping (in contrast to the DIP).''
> Before that, in the introduction, in the original and revised version, we have a paragraph devoted to the DIP explaining that Ulyanov et al. introduced the idea of using a deep neural network without learning as an image model.
>
> 2/ Regarding the theoretical contribution: We fully agree that a limitation of the theorem is that it pertains to a one layered version of the decoder. We are currently extending this to the multilayer case, but still have to address a technical difficulty in counting the number of different sign pattern matrices.
>
> Regarding the assumptions: The proposition uses the assumption that k^2 log(n_0)  / n <= 1/32. Here, the constant 1/32 is not optimal. k^2 is essentially the number of parameters of the model, and n is the output dimension.
> The proposition is only interesting if k^2 log(n_0)  / n <= 1/20 even without this assumption (due to the right hand side of the lower bound) therefore this assumption is not restrictive.
>
> The bound is applicable if the number of parameters, k^2 is smaller than a logarithmic term times the number of output parameters, i.e., it allows the number of parameters to scale almost linearly in the output dimension. This is the regime in which the deep decoder operates throughout the paper.
>
> We agree that many natural noise patterns have structure, and that those can be better approximated with deep models, and are thus more difficult to remove.
>
> 3/ We have added the sentence ``In the default architectures with $d=6$ and $k=64$ or $k=128$, we have that N = 25,536 (for k=64) and N = 100,224 (k=128)
> out of an RGB image space of dimensionality 512\times512\times3=786,432 parameters.'' to specify the number of parameters.
> Thanks for the suggestion to try second order method like LBFGS; we have tried LBFGS as a response to the reviewer's comment. It converges in significantly fewer iterations, but each iterations is so much more expensive that overall it optimizes slower than ADAM or gradient descent.
>
> Minor comments:
>
> 1/ Figure 4: We have added labels and the sentence ``Early stopping can mildly enhance the performance of DD; to see this note that in panel (a), the minimum is obtained at around 5000 iterations and not at 50,000.'' in the caption to clarify.
> Also, we have added the sentence ``Models are fitted independently for the noisy image, the noiseless image, and the noise.'', and rewrote the paragraph
> Thanks for pointing this out!
> We agree that here we present only results for one image, but we did carry out simulations for many images, and those plots are qualitatively the same for all the images considered. Thus our conclusions about the model do not only hold for one image.
>
> 2/ Normalization is applied channel wise. Let z{ij} be the j-th column in the i-th layer. Then z{ij} is normalized independently of any of the other channels.
>
> 3/ We have reworded the corresponding paragraphs to make clear that while we do not use convolutions, and thus this is not strictly speaking a convolutional neural network, it shares many structural similarities with a conventional neural network, as pointed out by the reviewer.
>
> 4/ The equation is correct in that the parameter choices in the paper are such that the deep decoder has much fewer model parameters N than its output dimension. Thus N is much less than n.
>
> 5/ We agree that it is not optimal to use unintroduced notation at this point, but we made this compromise so that we can illustrate the performance of the deep decoder without introducing its details, but wanted to give a reader the chance to later see exactly what parameters we used.
>
> 6/ Unfortunately choosing k=6 is too small to have a small representation error, i.e., to represent the image well. We have, however not hand-selected the 8 images shown out of the 64, and the other 64-8 images look very similar. We have all the images in the jupyter notebook that comes with the paper.
>
> 7/ Great question, it is faster to optimize the deep decoder since the adam/SGD steps are cheaper, but it indeed seems to require slightly more iterations for best performance than the DIP.

---

> > ### Comment · AnonReviewer1 · 2018-11-14
> > **Response**
> >
> > Thanks for addressing the first concern and other improvements, it is much better this way!
> >
> > I didn't get your response about the second concern though. Do you agree with my estimate that you would need no more than 2.5 parameters in the described case for the theorem to work (I could easily be wrong there)? If so, how is that "the regime in which the deep decoder operates throughout the paper"?
> >
> > I also want to share the following information with you without asking you to act on it or something, it's up to you to decide if it's important :)
> > I asked three colleagues to read the abstract and introduction (and nothing more) of the updated paper and 1 out of 3 thought that you are doing something completely different than DIP, i.e. different setup AND architecture. He was surprised to learn that it is not the case (which may indicate that my first concern is not fully addressed, but maybe it's fine, you can't expect everyone to understand everything).
> > And also, another 1 of the 3 thought that by non-convolutional you meant fully-connected.

---

> > > ### Author Response · Authors · 2018-11-17
> > > **response**
> > >
> > > Thanks for the feedback!
> > >
> > > Regarding the second concern, yes we do agree with the estimate. The theorem uses the assumption: k^2 log(n_0)  / n <= 1/32. This is equal to k^2 <= n/ (32 \log(n) ), and with the parameters of the described case, this indeed says that k^2 <= 2.56. However, the constant 1/32 is not optimal and can be made larger, e.g., by increasing the probability in the statement. Then the bound on k would be less restrictive. The condition essentially says that k^2, the number of parameters, should be smaller (by a logarithmic factor) than the dimension of the output. In this regime the decoder is underparameterized, and throughout the paper, we operate in the regime where the decoder is underparameterized.
> > >
> > > Thanks for asking your colleagues, that is certainly helpful! We do think that the architecture is quite different: our architecture is an underparameterized network without convolutions and an decoder-like structure, while the architecture of the DIP is overparameterized and has an encoder-decoder structure with skip connections. We think the setup is different in that we do not require regularization; however, if setup refers to fitting an un-trained model, then the setup is indeed the same.

---

> > > > ### Comment · AnonReviewer1 · 2018-12-03
> > > > **Updating the score**
> > > >
> > > > I see, thanks.
> > > >
> > > > I updated the score, thanks for the improvement in the paper!
> > > >
> > > > Note, however, that even though I liked the improvements, the discussion didn't feel very right to me (in particular the applicability of the theorem) and I was considering not updating the score because of that.

---

### Official Review · AnonReviewer2 · 2018-11-02
**A very interesting paper with good analysis and decent experiments..**

**Rating:** 8
**Confidence:** 4

**Review:**

Brief summary:

This paper presents a deep decoder model which given a target natural image and a random noise tensor learns to decode the noise tensor into the target image by a series of 1x1 convolutions, RELUs, layer wise normalizations and upsampling. The parameter of the convolution are fitted to each target image, where the source noise tensor is fixed. The method is shown to serve as a good model for natural image for a variety of image processing tasks such as denoising and compression.

Pros:
* an interesting model which is quite intriguing in its simplicity.
* good results and good analysis of the model
* mostly clear writing and presentation (few typos etc. nothing too serious).

Cons and comments:
* The author say explicitly that this is not a convolutional model because of the use of 1x1 convolutions. I disagree and I actually think this is important for two reasons. First, though these are 1x1 convolutions, because of the up-sampling operation and the layer wise normalizations the influence of each operation goes beyond the 1x1 support. Furthermore, and more importantly is the weight sharing scheme induced by this - using convolutions is a very natural choice for natural images (no pun intended) due to the translation invariant statistics of natural images. I doubt this would have worked so well hadn't it been modeled this way (not to mention this allows a small number of parameters).

* The upsampling analysis is interesting but it is only done on synthetic data - will the result hold for natural images as well? should be easy to try and will allow a better understanding of this choice. Natural images are only approximately piece-wise smooth after all.

* The use of the name "batch-norm" for the layer wise normalization is both wrong and misleading. This is just channel-wise normalization with some extra parameters - no need to call it this way (even if it's implemented with the same function) as there is no "batch".

* I would have loved to see actual analysis of the method's performance as a function of the noise standard deviation. Specifically, for a fixed k, how would performance increase or decrease, and vice versa - for a given noise level, how would k affect performance.

* The actual standard deviation of the noise is not mentioned in any of the experiments (as far as I could tell)

* What does the decoder produce when taking a trained C on a given image and changing the source noise tensor? I think that would shed light on what structures are learned and how they propagated in the image, possibly more than Figure 6 (which should really have something to compare to because it's not very informative out of context).

---

> ### Author Response · Authors · 2018-11-13
> **response**
>
> Many thanks for the detailed review!
>
> 1/ We agree that there are many elements of our architecture that are similar to that of a convolutional network, however the network does not perform convolutions. To reflect both points, we have revised the text to:
> ``The network does not use convolutions.
> Instead, the network does have pixelwise linear combinations of channels, and just like in a convolutional neural network the weights are
> are shared among spatial positions.
> Nonetheless, they are not convolutions because they provide no spatial coupling between pixels, despite how pixelwise linear combinations are sometimes called `1x1 convolutions.' '',
> and we have also added a subsection comparing the compression performance of our architecture to that of a decoder with convolution layers. In a sense, what the deep decoder is doing is separating multiple roles that proper convolutional layers fill:  the DD breaks apart the spatial coupling inherent to convolutions from their channel dependence and equivariance. Further, it says that the spatial coupling need not be learned or fit to data, and can be directly imposed by upsampling.
>
> 2/ Yes, the upsampling analysis in Figure 5 also extends to two-dimensional images. We agree that natural images are only approximately piece-wise smooth after all, and the deep decoder only provides an approximation of natural images (albeit a very good one).
>
> 3/ We agree and have changed `batch normalization' to `channel normalization' throughout.
>
> 4/ Great point; we have added the sentence ``The optimal $k$ trades off those two errors; larger noise levels require smaller values of $k$ (or some other form of regularization).
> If the noise is significantly larger, then the method requires either choosing $k$ smaller, or it requires another means of regularization, for example early stopping of the optimization.
> For example $k=64$ or $128$ performs best out of $\{32,64,128\}$, for a PSNR of around 20dB, while for a PSNR of about 14dB, $k=32$ performs best.''
>
> 5/ We do not mention the standard deviation, but do specify the SNR throughout (e.g., in table 1 in column identity). We have clarified this in the caption of the table.
>
> 6/ It essentially produces smooth noise then. The weights learned by the deep decoder pertain to the source noise tensor. We have added a corresponding figure to the jupyter notebook for reproducing Figure 6.

---

### Official Review · AnonReviewer3 · 2018-11-03
**A more principled DIP, interesting contribution.**

**Rating:** 8
**Confidence:** 4

**Review:**

In this paper, the authors propose a method for dimensionality reduction of image data. They provide a structured and deterministic function G that maps a set of parameters C to an image X = G(C). The number of parameters C is smaller than the number of free parameters in the image X, so this results in a predictive model that can be used for compression, denoising, inpainting, superresolution and other inverse problems.

The structure of G is as follows: starting with a small fixed, multichannel white noise image, linearly mix the channels, truncate the negative values to zero and upsample. This process is repeated multiple times and finally the output is squashed through a sigmoid function for the output to remain in the 0..1 range.

This approach makes sense and the model is indeed more principled than the one taken by Ulyanov et al. In fact, the DIP of Ulyanov et al. can hardly be considered "a model" (or a prior, for that matter), and instead should be considered "an algorithm", since it relies on the early stopping of a specific optimization algorithm. This means that we are not interested in the minimum of the cost function associated to the model, which contradicts the very concept of "cost function". If only global optimizers were available, DIP wouldn't work, showing its value is in the interplay of the "cost" function and a specific optimization algorithm. None of these problems exist with the presented approach.

The exposition is clear and the presented inverse problems as well as demonstrated performance are sufficient.

One thing that I missed while reading the paper is more comment on negative results. Did the authors tried any version of their model with convolutions or pooling and found it not to perform as well? Measuring the number of parameters when including pooling or convolutions can become tricky, was that part of the reason?

Minor:

"Regularizing by stopping early for regularization,"

In this paper "large compression ratios" means little compression, which I found confusing.

---

> ### Author Response · Authors · 2018-11-13
> **response**
>
> Many thanks for the review!
> Good point regarding the negative results; we have added a subsection in the revised paper entitled ``A non-convolutional network'', where we compare to a convolutional decoder and conclude that ``Our simulations indicate that, indeed, linear combinations, yield more concise representations, albeit not by a huge factor.''.
>
> Regarding the minor points, we have reworded the paragraph on regularizing, and changed `compression ratio' to `compression factor', and reworded such that `large compression factor' means large compression.

---

### Meta-Review · Area_Chair1 · 2018-12-12
**Simple model which achieves good results**

**Confidence:** 4
**Recommendation:** Accept (Poster)

**Metareview:**

In this work, the authors propose a simple, under parameterized network architecture which can fit natural images well, when fed with a fixed random input signal. This allows the model to be used for a number of tasks without requiring that the model be trained on a dataset. Further, unlike a recently proposed related method (DIP; [Ulyanov et al., 18]), the method does not require regularization such as early-stopping as with DIP.
The reviewers noted the simplicity and experimental validation, and were unanimous in recommending acceptance.